# Leveraging Structure Between Environments: Phylogenetic Regularization Incentivizes Disentangled Representations

Elliot I. Layne[1]          Dhanya Sridhar[2]          Jason Hartford[2]          Mathieu Blanchette[1]

[1]Computer Science Dept., McGill University, Quebec, Canada
[2]Mila - Quebec AI Institute, Université de Montréal, Quebec, Canada

## Abstract

Recently, learning invariant predictors across varying environments has been shown to improve the generalization of supervised learning methods. This line of investigation holds great potential for application to biological problem settings, where data is often naturally heterogeneous. Biological samples often originate from different distributions, or environments. However, in biological contexts, the standard "invariant prediction" setting may not completely fit: the optimal predictor may in fact vary across biological environments. There also exists strong domain knowledge about the relationships between environments, such as the evolutionary history of a set of species, or the differentiation process of cell types. Most work on generic invariant predictors have not assumed the existence of structured relationships between environments. However, this prior knowledge about environments themselves has already been shown to improve prediction through a particular form of regularization applied when learning a set of predictors. In this work, we empirically evaluate whether a regularization strategy that exploits environment-based prior information can be used to learn representations that better disentangle causal factors that generate observed data. We find evidence that these methods do in fact improve the disentanglement of latent embeddings. We also show a setting where these methods can leverage phylogenetic information to estimate the number of latent causal features.

## 1 INTRODUCTION

In computational biology, many problems feature samples that originate from differing, but related, environments. For example, gene expression data can vary significantly across different cell types, along with related downstream biological processes. Different cell types remain far from unrelated however, and these relationships can be described by a cellular differentiation tree, similar to a phylogenetic tree [Enver et al., 2005]. There has been a line of work in the causal representation learning field on developing methods capable of making predictions that are invariant across environments [Peters et al., 2016, Arjovsky et al., 2019]. Additionally, we note the ongoing line of work that seeks to leverage multiple environments for the purpose of causal discovery [Monti et al., 2020, Khemakhem et al., 2020]. In both cases, these methods are designed to be agnostic to the source of variation across environments, and as a result typically do not leverage prior knowledge about relationships between environments that is present in biology. In contrast, the bioinformatics literature has begun to develop methods specifically designed to learn from data samples originating from environments related by a phylogenetic tree [Layne et al., 2020]. Layne et al. demonstrated the ability to improve predictive performance by training a family of related functions, by using a regularization scheme that leverages the phylogenetic tree to ensure smooth changes between the functions associated to neighboring nodes, in a method they refer to as DendroNet.

Layne et al. focused on predictive performance, but prompts a natural question: does this regularization scheme also disentangle causal factors underlying the true data generative process? The intuition behind this hypothesis is similar to that motivating the use of meta-learning for disentangling causal structures as done by Bengio et al. [2020]. We consider that we are operating in a context where the effects of independent causal mechanisms varies gradually across environments, and that we are seeking to learn a family of functions $F$ that fit these environment-specific effects using as input some representation of the causal mechanisms. We presume that training these functions with disentangled representations of the causal mechanisms will be optimal for learning an ideal family of functions $F$, as opposed to using entangled representations as inputs. In our set-up, we

*Accepted for the 38th Conference on Uncertainty in Artificial Intelligence* (UAI 2022).

will justify that this optimal learning would correspond to learning a DendroNet model paying a minimized phylogenetic regularization penalty. We thus explore whether DendroNet's phylogenetic regularization scheme can in fact encourage learning disentangled representations of generative processes underlying observed data, via a series of simulation studies.

We note that in this work we are defining disentanglement of some embedding $e$ with respect to a vector of causal factors $z$ in the same manner as Eastwood and Williams [2018]. Informally, if $e$ is of greater dimension than $z$, we expect that each element $e_i$ should be informative of a single factor $z_i$. In the case that $e$ and $z$ are of the same dimension, a perfectly disentangled representation should recover the values of $z$, up to some permutation and linear transformations.

## 2  PROBLEM FORMULATION & METHODS

We assume we are operating in a multi-environment setting, with "environments" corresponding to different cell types that have diverged according to some evolution-like process, resulting in cell-type-specific distributions. This differentiation process imposes a natural tree structure that relates the joint distributions over different cell-types / environments: cells that are close together in the tree will have more similar distributions, while more distant relatives in the tree may experience larger distribution shifts.

We model this as follows: we assume there exist some set of $k$ different environments, $E = \{e_1, \ldots, e_k\}$, each of which is a unique leaf in a tree $T$. We typically assume that we cannot observe samples from internal nodes in the tree (which represent historical environments), but our framework is agnostic to this. The environments each define a different joint distribution, $P^{(e_i)}(X, Z, Y)$ over some observed features $X \in \mathbb{R}^d$, targets $Y$, and a latent vector $Z \in \mathbb{R}^k$. We assume that, at tree node $i$ with environment $e_i$, the latent variables $Z$ are some function, $\Phi(\cdot)$ of the observed features and an independent noise variable, $u_z$, such that $Z = \Phi(X, u_z)$. We further assume the target variable $Y$ is some function, $\Psi(\cdot)$, of the latent variables $Z$, and a noise variable, $u_Z(T)$, which induces dependence across environments $Y = \Psi(Z, u_{(e_i)}(T))$. Across all environments, we assume that $\Phi(\cdot)$ does not vary. However, the conditional distribution $P^e(Y|Z)$ induced by $\Psi(\cdot)$ is subject to evolution across environments because of the changing noise variables. The difference in the distributions $\Psi_{(e_i)}(Z)$ and $\Psi_{(e_j)}(Z)$ is assumed to be proportional to the distance between $(e_i, e_j)$ in tree $T$ such that,

$$d_T(e_i, e_j) \leq d_T(e_i, e_k) \implies \\ KL(\Psi_{(e_i)}, \Psi_{(e_j)}) \leq KL(\Psi_{(e_i)}, \Psi_{(e_k)}) \tag{1}$$

Where $d_T(a, b)$ denotes the distance between nodes $a$ and $b$ in tree $T$, and $KL$ refers to the Kullback–Leibler divergence.

**Parametric Example of Environment Evolution**  A particular choice for $\Psi(Z)$ is a linear map with parameters $w$, and a Gaussian change in the values of $w$ across each parent-child pair of environments $(e_{parent}, e_{child})$. We denote the difference in $w$ between a parent environment $e_i$ and child environment $e_j$ as $\delta_j$. We can calculate the likelihood of some child environment $e_{child}$ parameters $w_{child}$ using a Gaussian distribution centered at the parent environment:

$$w_{child} \sim \mathcal{N}(w_{parent}, \sigma^2) \tag{2}$$

The variance parameter $\sigma^2$ in Equation 2 depends on the rate of the evolutionary process between environments. In general, it is assumed that evolution happens slowly, and that large values of $\delta$ are unlikely to be observed.

### 2.1  DENDRONET & PHYLOGENETIC REGULARIZATION

This work further interrogates the DendroNet method and analyses its behaviour in our multi-environment problem setting. We seek to determine under what conditions, if any, the ability to account for structure between environments via a phylogenetic regularization scheme can have properties that are beneficial from a causal representation learning perspective.

As described by Layne et al. [2020], DendroNet addresses the problem of learning from non-i.i.d. data resulting from evolution between environments, such as species or cell types. This is accomplished by training a family of related functions for the prediction task at hand. The implementation is as described in Figure 1: a set of weights is initialized representing parameters located at the root of a phylogenetic tree. Each edge in the tree contains a set of trainable offsets for the root parameters, referred to as edge-specific $\delta$ parameters. Tree-based phylogenetic regularization addresses overfitting by ensuring smooth differences between functions related to different samples in the training data, which is desirable as evolution is typically a gradual process.

The loss function when training a DendroNet model includes both the prediction-based loss (e.g. MSE or cross-entropy) as well as regularization penalties controlling the amount of edge-specific updates $\delta$ to the model parameters. For example, the loss corresponding to the model in Figure 1 could be as follows for a regression problem, where $\lambda$ is a scaling parameter, referred to hereafter as the delta penalty factor (DPF):

$$\text{Loss} = \sum_{u \in \{A,B,C\}} (\hat{y_u} - y_u)^2 + \sum_{u \in \{A,B,C,D\}} \lambda |\delta_u|_2^2 \tag{3}$$

We reiterate the central intuition motivating this work: in a setting where the effects of underlying generative factors are

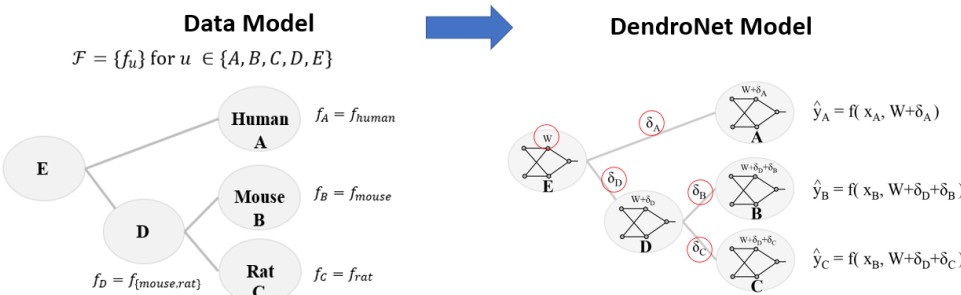

Figure 1: A DendroNet model for a dataset with samples originating from three different species. A set of weights $W$ is associated to the root. A trainable update to $W$ is associated to each edge. The prediction function $f_u$ at node $u$ will make predictions on data samples originating from species A. Trainable parameters are highlighted in red. Adapted from Layne et al. [2020].

gradually evolving across a set of environments, DendroNet-style models should be incentivized to learn representations that *disentangle* these generative factors, as this should facilitate achieving optimal predictive performance while paying a minimal cost in terms of DPF penalty.

While DendroNet was motivated by the setting where samples in a dataset of interest could originate from different species (i.e. different environments), the approach is valid in other settings where sub-groups have some sort of hierarchical dependency structure, such as a collection of cell-types derived from stems cells via a cellular differentiation tree.

## 3 EXPERIMENTATION

We empirically investigate the capability of DendroNet to learn representations that disentangle latent factors whose effects vary across a set of related environments. As biological motivation, we posit a simulation where the set of related environments are different cell types.

Briefly, in our simulation-based results we found that:

1. DendroNet was capable of making accurate predictions across varying environments.

2. DendroNet learned partially disentangled representations of the latent causal factors, and could be used to infer the dimensionality of the latent space.

3. In the case of a model mis-specification of the data generating process, we found evidence that variation between environments improved the disentanglement performance.

### 3.1 SIMULATED DATA GENERATING PROCESS

We start by defining some notation for our simulation:

- A set of environments, each representing a different cell-type $c \in C$.

- Tree $T$ relating each of the cell types. In this study, we used a simple caterpillar tree with five leaves.
- Collection of feature vectors $X^{d,n}$. Biologically, these represent observable gene expression values of $d$ genes collected from a population of $n$ different cell types.
- Vector of target phenotypes $Y^n$.
- Collection of latent factors $Z^{k,n}$. Biologically, these represent the direct causal factors influencing $Y$.
- We simulate a mapping: $Z^{n,k} = \Phi(X^{n,d})$
- Each $X_i \in X$ is assigned to one of the cell types.
- Each cell-type $c$ has a vector of weights $w^c$. Phenotypes in $Y$ are generated with the cell-type specific process: $Y_i^{(c)} = Z_i \cdot w^c + \varepsilon$, where $\varepsilon$ is a small amount of random noise

The interactions among all of these components are summarized in Figure 2. The values of the weights $w^c$ are drawn from a multivariate Gaussian. The covariance matrix relating values of $w$ for each cell type scales the correlation between some minimum value (specific to each experiment, discussed below) and 1, proportional to the inverse of the patristic distance between the two cell-types in the tree.

Unless otherwise noted, all experiments used a tree of 5 cell types with 5000 simulated samples each, split into 50% training, 25% validation and 25% test. The dimensionality of $X$ and $Z$ were varied across experiments.

### 3.2 EXPERIMENTAL DETAILS

The architecture for the models mirrored much of the simulated data generative process. All experiments were conducted using a two-component model. The first component in all experiments was an encoder, referred to hereafter as component $E$. The encoder $E$, given a desired embedding size $m$, took as input samples of gene expression values $X$ and produced as output an embedding $E(X)$. $E$ was implemented as: $E(X) = \tanh(AX)$, where $A_{d \times m}$ is a trainable

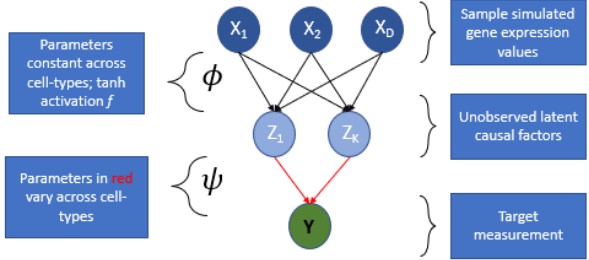

Figure 2: An example of the simulated data generative process. The first row represents components of $X$, the second row components of $Z$ containing the latent causal factors, and the final row $Y$. Connections between $X$ and $Z$ (which make up the mapping $\Phi$) are passed through a tanh activation function. The weights $w$ in red are the cell-type specific components, which define $\Psi$. Small amounts of random Gaussian noise $\varepsilon$ are added to all values of $Y$.

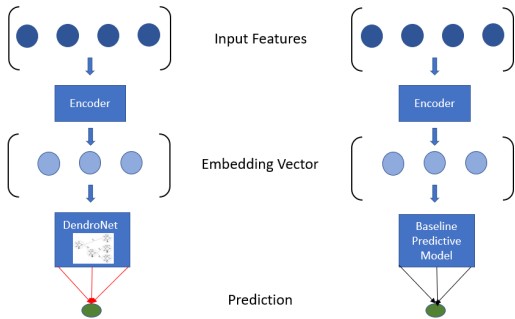

Figure 3: The two model architectures used in this study. On the left, the encoder is jointly trained with a DendroNet model that will fit an adaptive set of linear weights specific to each cell type. On the right, the encoder is jointly trained with a traditional phylogeny-unaware set of linear weights.

matrix of weights. The value of $m$ varied by experiment, as specified further below. $E$ was given no information about cell types.

The encoder was always jointly trained with a predictive model, referred to as $P$. $P$ used the embeddings from $E$ as input, and output a prediction for phenotype $Y$. $P$ was a simple linear model: $P(Z) = B \cdot Z + b$. In our baseline model, a single weight vector $B$ was used across all environments. In our DendroNet model, $B$ was allowed to vary along the branches of the tree. In the DendroNet case, the list of cell-type assignments was required for all samples in $X$, in order to train and apply the optimal cell-type specific adaptations.

It should be noted that while the primary metric of interest was the ability of $E$ to capture disentangled representations of causal elements in $Z$, the only lever to encourage this was the prediction loss from $P$. No explicit regularization terms encouraging disentanglement were placed on $E$ or its outputs.

All models were implemented using the Pytorch library Paszke et al. [2019]. The analysis involving DCI scores used an implementation adapted from the version produced by Locatello et al. [2019]. Hyper-parameter values were tuned on a validation set, with a typical setting for the DPF value to be 0.01.

### 3.3 RESULTS

Throughout experimentation, we examined the effects of changing the amount of variation between environments, as seen along the x-axis in Figures 4 and 5. As described in section 3.1, this was achieved by manipulating the covariance matrix when sampling the parameters $w^c$ specific to each cell type, which is equivalent to increasing or decreasing the variance parameter described in equation 2. In a slight

abuse of notation, we refer to this as the level of correlation between causal factors across cell types. This should not be confused with the introduction of correlation between different elements of $w$ within a single cell type.

Our first result demonstrated the ability of the DendroNet method to fit phylogenetically distributed data, using our simulated data generative process. We used an observable gene expression vector $X$ of dimension $d = 10$, and both latent causal vector and encoder $E$ embedding of dimension $k = 5$. We compared the average MSE of both a DendroNet and baseline predictive model $P$, while varying the amount of correlation between cell type specific weights applied to $Z$. The results are displayed in Figure 4. At all levels of correlation between cell type specific weights, DendroNet performed near optimally, with loss approximately equal to the random noise injected into the target values. The baseline method also had loss approaching noise levels when there was no variance across cell types. However, its performance degraded quickly as the amount of variation between cell types increased.

We then compared the ability of encoder $E$ to produce embeddings that disentangle the unobserved latent factors generating $Y$ in the simulation process, when trained jointly with different downstream predictors $P$ (either DendroNet or the baseline model). We measured disentanglement using the DCI scores between the embeddings output by $E$ and the true latent $Z$ vectors. We set $d = 16$, $k = 5$, and the encoders produced embeddings of dimension 8. We varied the amount of correlation between the cell-type specific parameters in the generative process. Results are depicted in Figure 5, and discussed in more detail in Section 4. In brief, DendroNet achieved superior DCI scores across all experimental settings.

We then analyzed the ability of DendroNet to identify the true dimension of $Z$. We repeatedly trained the encoder-predictor model with varying settings of size for the em-

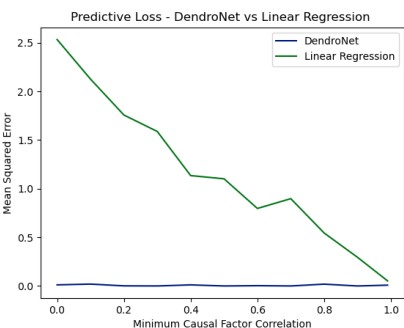

Figure 4: A comparison of the MSE on the test set across DendroNet and the baseline method, averaged across 25 replicate experiments. With any amount of variation allowed between causal factors in different cell types, the baseline method experienced higher error than DendroNet. DendroNet achieved loss approximately equal to the amount of random noise injected into the target values at all levels of correlation between causal factors.

bedding produced by the encoder $E$, as well as the size of causal latent factors $Z$. In all cases, we used a feature vector of simulated gene expression values of size 10. A moderate amount of variation was permitted across cell-type specific weights. The results are displayed in Figure 6. The loss obtained by DendroNet was elevated for embedding sizes less than the dimension of $Z$, and then became roughly equal to the amount of random noise for all settings of embedding size equal or greater to $Z$. In the case of the baseline model, which was not capable of capturing the variations in causal factors across cell type, improvements in loss continued as the embedding size increased past the dimension of $Z$.

## 4    DISCUSSION

The results in Figure 5 support the hypothesis that the use of phlyogenetic regularization can incentivize learning disentangled data representations. In the top sub-figure, we can see that DendroNet achieved a clear improvement over the phylogeny-unaware comparison method with regards to DCI scores. While both methods performed very well when there was no variation between cell types, the performance of the baseline method degraded quickly as the amount of cross-cell variation increased, while DendroNet maintained near optimal performance. In the bottom sub-figure, we see the performance of the two methods when there was a mismatch between the embedding architectures and data generative process. In this more challenging setting, where neither model could maximize the DCI scores, the advantage of the phylogenetic regularization became more clear. When there was no variation between cell types, the DendroNet and baseline methods achieved similarly low DCI scores. When variation between the cell-type specific

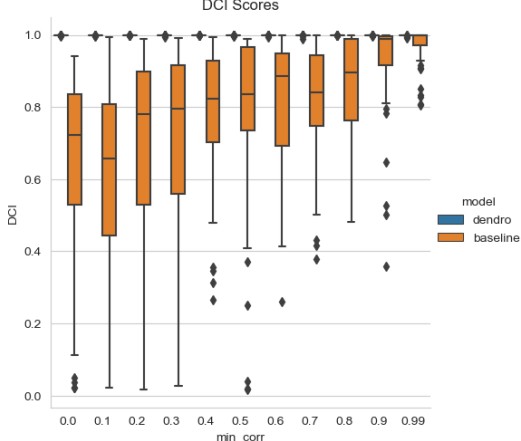

(a) Model matches generative process

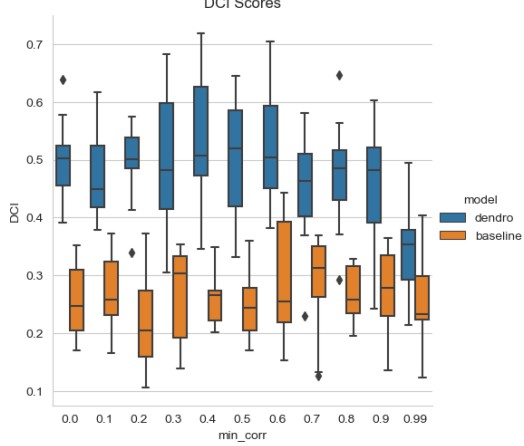

(b) Model mis-specification

Figure 5: Top: DCI scores when the encoders were capable of expressing $\Phi$ exactly. On the y-axis, the DCI scores between the outputs from $\Phi$ and the learned embeddings. On the x-axis, the minimum correlation between the least related pair of environments, used when generating the cell-type specific weights mapping $Z$ to $Y$. DendroNet scores are tightly grouped at the top of the y-axis. Bottom: the same comparison between DendroNet and the baseline model, but with a slight mis-specification in the models. The generative process had a fully linear map $\Phi$, while the embeddings were produced using a tanh activation function.

parameters was introduced, despite this making the classification task more challenging, the DCI scores improved. Presumably, this is because the variation created an incentive for DendroNet to learn disentangled embeddings, so that it could model the variations between cell types while paying a minimal DPF penalty.

In Figure 6 we demonstrated the ability of DendroNet to estimate the true number of independent causal factors, without the need to directly observe them. We supposed

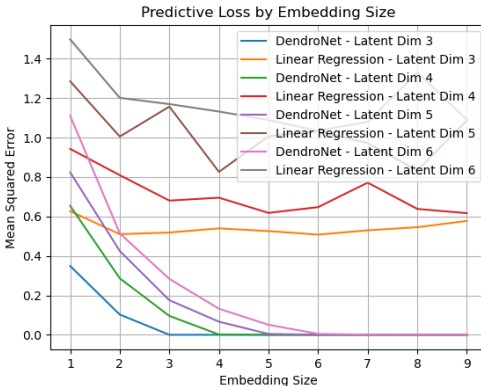

Figure 6: A comparison of the MSE observed by DendroNet and baseline methods as the embedding size of the encoder portion of the model were varied. In all cases, the true data generative process used input features of dimension 10, with moderate variation in the causal factors in $Z$ across cell types (minimum correlation across cell-types fixed to 0.6). The dimension of $Z$ was varied from 3 to 6. In all cases, as the size of the encoder output increased, the loss for DendroNet decreased rapidly until embedding dimension was equal to the true dimension of $Z$, and then plateaued. Loss for the baseline model did not display this trend; the minimum loss was achieved with embedding sizes that are not equal in dimension to the true $Z$.

that an ideally trained DendroNet model should be able to achieve optimal predictive performance once the embedding produced by encoder $E$ became equal in size to the latent causal vector. Any further increase in embedding size should offer no benefit, which is indeed what was observed. An interesting future direction would be to establish sufficient conditions for this phenomenon to hold, so that we may use this process of re-training with disentangled embeddings of different sizes to determine the number of causal factors influencing a biological trait of interest, such as determining the number of parents that a given gene has present in a gene interaction network (or, in more abstract terms, the number of parents for a node in a structural causal graph).

In this work, both the simulated generative process, and the mappings from embedding to phenotype were linear, meaning that DendroNet was capable of expressing the true generative function, which may have aided with the recovery of the true number of causal factors. In general, the true effects of gene expressions (or other biological traits) on target measurements are much more complex and non-linear. We leave to future work the investigation of the disentanglement behaviour of the DendroNet method in this setting.

In summary, we have explored the ability of phylogeny-aware machine-learning methods to learn data representations that disentangle causal factors in the underlying data

generating process. We motivated our investigation with relevant biological questions concerning patterns of variation across cell types, and performed simulation studies. Empirical results showed evidence that the DendroNet method encourages disentangled representations, by leveraging phylogenetically structured variation between environments. We also presented evidence that DendroNet training regimes can aid in identifying the number of latent causal factors relevant to a target variable.

Future work should seek to build on the these findings, and further utilize phylogenetic information in making improvements to causal representation learning and causal discovery methods.

## Author Contributions

MB conceived of the initial hypothesis, and gave guidance throughout the experimental process. EL developed the initial experimental design, implemented the code, created the figures and wrote the initial draft of the manuscript. DS and JH both made contributions in further developing the problem formulation, and improving experimental design. All authors contributed to editing the manuscript.

## Acknowledgements

We acknowledge the support of the Natural Sciences and Engineering Research Council of Canada (NSERC).

Nous remercions le Conseil de recherches en sciences naturelles et en génie du Canada (CRSNG) de son soutien.

The authors would like to thank Paul Bertin, Alex Tong and Yue Li for helpful discussions.

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

# A  APPENDIX

## A.1  DCI SCORE

We define the disentanglement score $D$ for an element $i$ in an embedding as in Eastwood and Williams [2018]: $D_i = 1 - H_K(P_i)$, with $H_K(P_i) = -\sum_{k=1}^{K-1} P_{ik} \log_K$ being the entropy. $P_{ij}$ denotes the probability that element $i$ in the embedding is important for predicting $z_j$. Thus disentanglement for an embedding entry $e_i$ is maximized (to 1.0) when it is significant for only a single entry $z_j$, and is equal to 0 when it is equally likely to be predictive of all entries in $z$.