# OpenReview forum: "Leveraging Structure Between Environments: Phylogenetic Regularization Incentivizes Disentangled Representations"
_auai.org/UAI/2022/Workshop/CRL — CRL@UAI 2022 Poster_

### Official Review · Reviewer_6GiY · 2022-06-28
**Good paper with interesting ideas**

**Rating:** 7
**Confidence:** 4

**Review:**

I enjoyed reading the paper and found the overall idea to be well thought out and intuitive. Moreover, I congratulate the authors on presenting methods directly motivated by practical challenges (in this case gene expression data). I think this submission would be significantly improved if the authors also presented some real world data results together with their (extensive) simulation results.

Moreover, I think the overall presentation of the methods could be improved to make it clear what the contributions are. My understanding is that DendroNet introduces a hypernetwork style architecture, where there is a shared set of parameters, $W$, and then each environment $e$ is allow to introduce its own additional terms,  $\delta_e$. These additional terms can operate directly over $W$ or in some hierarchical fashion (it is unclear if the associated tree structure is provided apriori or somehow inferred from the data).

Minor comments
1. The text in FIgure 1 is too small to read without aggressively zooming in.
2. The paper, although well written, misses a large number of relevant citations, particularly from the causality literature. For example Peters et al (2016) provide identifiability proofs of invariant predictors in multiple environments. Similarly, Monti et al (2019) and Khemakhem et al (2020 leverage multiple environments. Also missing is a discussion as to why DendroNet is so much better than ERM, especially with ERM has been shown to be a strong baseline when properly tuned (Gulrajani & Lopez-Paz, 2020).


Ref
Peters, J., Bühlmann, P. and Meinshausen, N., 2016. Causal inference by using invariant prediction: identification and confidence intervals. Journal of the Royal Statistical Society: Series B (Statistical Methodology), 78(5), pp.947-1012.

Monti, R.P., Zhang, K. and Hyvärinen, A., 2020, August. Causal discovery with general non-linear relationships using non-linear ica. In Uncertainty in artificial intelligence (pp. 186-195). PMLR.

Khemakhem, I., Kingma, D., Monti, R. and Hyvarinen, A., 2020, June. Variational autoencoders and nonlinear ica: A unifying framework. In International Conference on Artificial Intelligence and Statistics (pp. 2207-2217). PMLR.

Gulrajani, I. and Lopez-Paz, D., 2020. In search of lost domain generalization. arXiv preprint arXiv:2007.01434.

---

### Official Review · Reviewer_cCx5 · 2022-06-28
**The paper leverages existing hierarchy of environments to better learn disentangled causal factors generating the data but could use minor revisions.**

**Rating:** 7
**Confidence:** 2

**Review:**

The paper leverages existing hierarchy of environments to better learn disentangled causal factors generating the data with clear applications to computational biology where the difference between environments could correspond to different cells along a differentiation tree or species over a phylogenetic tree. The problem is interesting and well motivated. The solution proposed makes sense. Models for close environments on the phylogenetic tree are biased through the proposed regularization to be close in the weight space.

Some minor comments:
1. The DCI score is not defined in the paper, it would be beneficial to include its definition (either in appendix or in main text) to make the manuscript self-contained.
2. Some issues with figures: Figure 1 is too tiny. Figure 5 (a) does not show the dendronet while the discussion text makes reference to it.

---

### Official Review · Reviewer_3rR3 · 2022-06-29
**Interesting work on analysis of disentangled representations in structure aware learning, will benefit from clearer evaluation**

**Rating:** 7
**Confidence:** 3

**Review:**

_**Contributions**_

This paper sheds new light on the ability of structure-aware Dendronet models to disentangle underlying true causal factors of variation for predicting the cell phenotype. Inspired by Bengio’s viewpoint on meta learning objectives to promote disentanglement, the authors draw a natural parallel between the 2 settings. They probe whether the Dendronet model (which relies on phylogenetic information of evolution of cell types i.e. a prior structure between environments) can learn embeddings which disentangle causal factors. This probation is done via a simulation study where embeddings are learnt via an encoder and a predictive model. Two kinds of predictive models are used for evaluation and compared via the metrics of Mean Squared Error (MSE) and DCI. They also throw light on how the true dimension of the true casual latent vector can be recovered via the Dendronet predictive model.

_**Strengths**_

Overall, the paper is very well written and motivated. The authors do a good job at explaining the setup which may not be obvious for someone not familiar with computational biology. The connection with Bengio's work on meta learning is intuitive. The study is well evaluated, showing prospective ways of analysing disentangling of representations in other application areas. The study on recovering the latent dimension is particularly interesting.

_**Main Concerns**_

1. It is unclear whether the improvement w.r.t the baseline is due to disentanglement, or simply an artefact of the extra number of parameters in the Dendronet model? It looks like weight vectors for each cell type can be varied in Dendronet as compared to the baseline, which naturally increases the capacity of Dendronet to learn more complex relations. Perhaps, is there any oracle which serves as an upper bound for the performance?
2. An intuitive explanation of the "ideal performance" is lacking. The analysis is done via increasing the amount of allowed variation between different cell types - why is this a valid parameter to vary for tracking the MSE and DCI scores?
3. How does the mis-specification study contribute to the concept of disentanglement?

_**Other questions and suggestions**_

It would be nice if the authors can incorporate these questions into the next iteration of the paper:

1. Overall, the experimental section can benefit from a reorganization. Currently,  the reader needs to keep jumping to the figure caption to know the conclusion. A one line conclusion of the results during the setup would greatly help in coherence.
2. The DCI score may not be widely known and should be explained briefly - for e.g., what kind of values are better and what is the expected trend for an ideal oracle?
3. Figure 6 is hard to track because of the variety of colours - a better representation scheme could be chosen.
4. Why is the noise kept constant for all nodes?
5. Are there any real life datasets on which this evaluation can be conducted?
6. It would be nice to see some typical values of the regularization constant (DPF) used in the experiments.

---

### Meta-Review · Program_Chairs · 2022-07-06

**Recommendation:** Accept (Poster)
**Confidence:** 4

**Metareview:**

All reviewers praised the paper for introducing an interesting idea motivated by a clear application to computational biology.

---

### Decision · Program_Chairs · 2022-07-06

Accept (Poster)